# The Optimized Energy Saving of a Refrigerating Chamber

**Whei-Min Lin [1], Chung-Yuen Yang [1] , Ming-Tang Tsai [2],\*  and Hong-Jey Gow [3]**

[1]   Department of Electrical Engineering, National Sun Yat-Sen University, Kaohsiung 807, Taiwan;
      wmlin@mail.nsysu.edu.tw (W.-M.L.); km9315127@gmail.com (C.-Y.Y.)
[2]   Department of Electrical Engineering, Cheng-Shiu University, Kaohsiung 833, Taiwan
[3]   R&D, Kuen-Ling Machinery Refrigerating Co., Ltd., Kaohsiung 82644, Taiwan; ghjey19@kuenling.com.tw
\*   Correspondence: k0217@gcloud.csu.edu.tw; Tel.: +886-7-7310606

**Abstract:** This paper proposes a control strategy for the energy saving of refrigerating chambers. Combining binary coding and proteome reorganization, the binary proteome algorithm (BPA) is proposed to solve this problem. The refrigeration system model is firstly established based on the performance data of compressors and temperature measurements of each refrigerating chamber. The objective function is an averaged coefficient of performance (*COP*), which considers the switching loss of the compressors, power consumption of the compressors, and refrigerating capacity of the chambers. The control strategy is defined as an optimization problem with constraints to avoid the ineffective operation of a refrigeration system for improving the *COP*. BPA is adopted to solve the control strategy for optimizing energy saving. The effectiveness and efficiency of the BPA are demonstrated using a real system, and the results are compared with the original control strategy. Results show that the average power consumption drops from 115.92 kW to 108.82 kW, and the average *COP* value rises from 1.92 to 2.03. The proposed control strategy is feasible, robust, and more effective in energy-saving problems. Other than energy saving, the proposed control strategy also has the benefits of reducing the evaporator frost formation, which allows the products to avoid chill damage.

**Keywords:** refrigerating chamber; binary coding; proteome; coefficient of performance

## 1. Introduction

The refrigeration system is closely related to human life, and modern people need refrigeration equipment for preserving the freshness and delicacy of food. From savage to civilized times, food preservation is closely related to human existence. With the continuous advances of civilization, science, and technology, the "cold" not only preserves the freshness of food, it also enables people to enjoy the foods of different seasons and different regions. A refrigerating chamber is a space with a refrigeration system and heat insulation walls [1]. The refrigerated storage of fish began in about 1861, the refrigerated preservation of poultry meat began in about 1870, the refrigerated preservation of shellfish, shrimps, and eggs began in about 1890, the cold storage of fruit, ice cream, and fruit juice began in about 1904, and the last refrigeration technique for preserving food was the freezing of vegetables, which began in about 1920. Therefore, the refrigerating chamber was developed for over 150 years [2]. As food can be used more effectively following low-temperature food preservation, food transport costs can be substantially reduced. Therefore, the refrigerating chamber will continue to be an irreplaceable technology in the future. How to use and operate refrigerating chambers more efficiently is being actively studied by industrial and academic circles [3].

In recent years, according to the estimation of various countries' agricultural organizations, some developing countries have as much as 50% of food spoiled in transportation, which is due to the lack of refrigerating equipment; thus, the United Nations (UN) encourages installing cold stores [4]. Taiwan's living standards continuously improved in recent years, and, in order to supply fresh and diversified foods for people, cold stores are already an indispensable piece of equipment. Refrigeration systems consume a high percentage of electricity in various areas, including home power consumption, service industries, and industrial processes, and electric power is the major cost of cold storage. According to the statistics of South Korea, the electric power cost of cold storage suppliers accounts for 15–20% of their total operating cost [5]. Therefore, how to save the energy consumption of refrigeration systems, while simultaneously maintaining the quality of frozen food, is now the general objective of refrigeration systems.

The conventional control techniques of a refrigeration system, such as PID controls [6] and on/off controls [7], highly depend on the operation of experienced operators [6,7]. Salazar et al. [6] developed a PID control for a single-stage transcritical $CO_2$ refrigeration cycle, and Li et al. [7] proposed an optimal on/off control of refrigerated transport systems for minimizing energy consumption. With the advance of science and technology in the past decade, advanced control methods, such as adaptive control [8] and intelligent control [9], can be implemented in refrigeration systems, further enhancing efficiency and quality. Powell et al. [10] set up a novel technique to solve dynamic chiller loading in a district cooling system with thermal energy storage. In the research area of heating, ventilation, and air conditioning (HVAC), many scholars studied the optimal design, operation, and control method of water chillers, air supply systems, and ice thermal storage systems, in order to provide the maximum benefit of equipment for energy savings and comfort [11]. Cutillas et al. [12] offered an optimum design and operation of an HVAC cooling tower to implement a maximizing control strategy in order to reduce both energy and water consumption. The ice storage system runs the refrigerating compressor at off-peak loads during the night. The water is frozen to store plenty of latent heat via phase change, and it is then discharged to reduce the load profile of air conditioning during peak load or semi-peak load [13].

An analysis of a refrigerated vehicle for food transport was developed in Reference [14], and Hoang et al. [15] used a heat transfer model of a refrigerated vehicle in a ventilated cavity loaded with food products. By using a cascaded robust temperature control, the temperature control of a refrigerated chamber was proposed to generate the required cooling power as efficiently as possible [16]. Oludaisi et al. [17] proposed the best approach to minimize energy consumption in refrigerated vehicles through an alternative external wall. Reference [18] was devoted to the determination of the efficiency of air curtain units applied to heat and moisture insulation of refrigerated cambers. The results indicated that these researches can be effectively applied to obtain optimal operation for refrigerated cambers. The objective of the optimized energy saving for refrigerating chambers is a non-linear integer programming problem with different operating limitations. With non-linearity and discrete nature considered in the objective function, it is difficult to design a quick and optimal procedure using only one algorithm. Therefore, the binary proteome algorithm (BPA) is proposed to solve this problem.

This paper proposes a binary proteome algorithm (BPA) that combines binary coding [19,20] and proteome reorganization [21]. Binary coding is used in BPA to search for the optimal on/off status of the refrigerating chamber. Bionics is a competition mechanism that was implemented to automatically determine the choice of either status. BPA has three advantages: (1) complicated problems are solvable, (2) it has better performance than other algorithms, and (3) there is a greater likelihood of achieving a global optimum versus heuristic methods. This paper focuses on the minimization of coefficient of performance (*COP*) [22], which considers the switching loss of the compressors, power consumption of the compressors, and refrigerating capacity of the chambers. A real system is demonstrated to verify the effectiveness and efficiency of the BPA. Results show that the proposed control strategy is feasible, robust, and more effective in energy-saving problems.

## 2. System Structure

The refrigeration cycle system comprises four major components: (1) compressor, (2) condenser, (3) expansion valve, and (4) evaporator, as shown in Figure 1. The compressor, which is the heart of the system, establishes the high/low voltage of the system, and is the foremost power consumption source, as it compresses low-pressure vaporous refrigerant into high-pressure vaporous refrigerant to enter the condenser. The condenser discharges the heat of the refrigerant to the outside, so that the high-pressure vaporous refrigerant is liquefied to enter the expansion valve. The expansion valve expands the high-pressure liquid refrigerant into low-pressure liquid to enter the evaporator. The refrigerant temperature of the evaporator is lower than the temperature of the refrigerated space; thus, the refrigerant absorbs the heat of refrigerated space, and evaporates into the vapor status to enter the compressor, thereby forming a cycle.

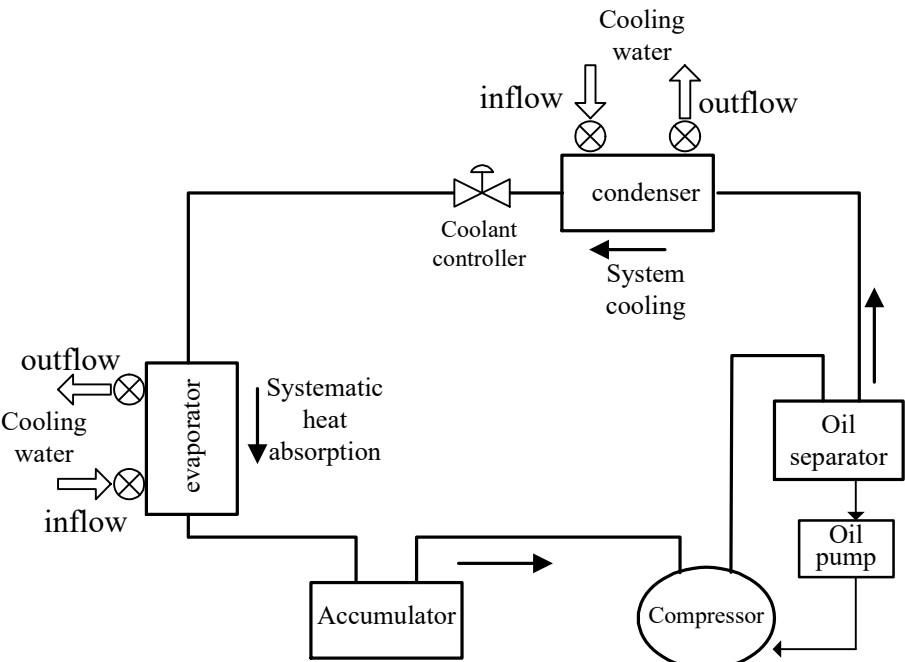

**Figure 1.** The refrigeration cycle system.

Two reciprocating compressors of the same capacity are connected in parallel in this study, providing refrigeration for 10 refrigerating chambers, as shown in Figure 2. The number of compressors cutting into operation can be controlled according to the changes in the refrigeration load to control the capacity of the refrigeration system. The refrigeration system supplies refrigeration for 10 refrigerating chambers, where the temperature of each refrigerating chamber must be controlled between the ceiling temperature (−10 °C) and the floor temperature (−20 °C). The conventional control method is "two-point temperature control"; when the temperature of a refrigerating chamber is higher than the ceiling temperature, the controller turns on the refrigerant control electromagnetic valve of the refrigerating chamber (on), the low-temperature refrigerant flows into the evaporator in the chamber to supply cold for the refrigerating chamber, and the temperature begins to drop. When the temperature is lower than the floor temperature, the controller turns off the electromagnetic valve (off), and the temperature rises slowly; thus, the refrigerating chamber temperature reciprocates between the two temperature points. The controller must simultaneously control the number of compressors cutting into operation. The controller calculates the total number of refrigerating chambers for cold supply, i.e., the number of electromagnetic valves in the on status, and this number is proportional to the refrigeration load, in order to determine the number of compressors cutting into operation.

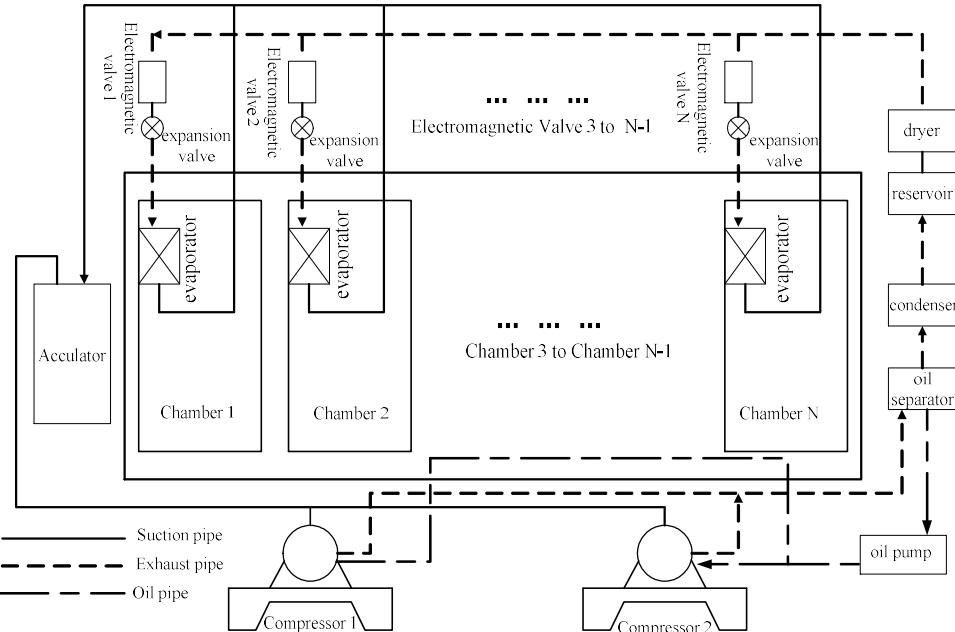

**Figure 2.** Refrigerated chamber.

This study proposes a control strategy for refrigeration system energy-saving control, which maintains the system at a high coefficient of performance (*COP*). Therefore, the controller is designed to flexibly switch the on/off status of refrigerating chambers, meaning the status of the electromagnetic valve is changed by the controller before the refrigerating chamber temperature exceeds the upper or lower temperature limit, which regulates the refrigeration load of the system, such that the refrigeration system runs at high *COP* for a long time; however, the system cycling loss and abrasion loss of machine members must be considered. Therefore, as the electromagnetic valve should not be switched too frequently, a criterion must be established to evaluate the benefit of switching the status of the electromagnetic valve by the controller, in order to save energy. The energy-saving control strategy is a control problem of a hybrid system, including the continuous status variable and discrete status variable (e.g., temperature and status of electromagnetic valve), which is a mixed-integer programming problem. It is difficult to determine the theoretically optimal solution to this type of problem, as the optimal solution is obtained mostly by searching all the feasible solution spaces, and dynamic programming or the branch and bound method is usually used to reduce the search space [23,24]. For a specific problem, an efficient heuristic search method or an artificial intelligence algorithm must be designed according to the characteristics of the problem. This paper proposes the binary proteome algorithm (BPA) to maintain the system at high *COP*.

## 3. The Control Strategy for the Energy Saving of a Refrigerating Chamber

### 3.1. Proteome Reorganization

This paper proposes the BPA to reduce the electricity costs of a refrigerating chamber, which can assist suppliers to rapidly and accurately perform the most efficient operation of a refrigerating chamber. The protein is composed of a long chain of amino acids, and, as there are 20 kinds of amino acids, different proteins can be synthesized [25,26]. Many proteins with different functions are synthesized according to the different arrangements, folding modes, quantities, and shapes of amino acids. In the course of protein synthesis, the messenger RNA (mRNA) completed by transcription moves from the nucleus to the ribosome, which is led by ribosomal RNA (rRNA). Afterward, the transfer RNA (tRNA) comes together to perform translation; the ribosome reads the gene on the mRNA and determines the amino acid required by the gene, and the tRNA combines with the corresponding mRNA gene. Then, the tRNA retains the amino acid, while the tRNA leaves. As the tRNA goes in and out of the

ribosome continuously, the amino acids are linked continuously, while the mRNA and tRNA pair to form a long-chain substance. However, as the long chain of amino acids is not steady, the synthesized amino acids fold and bend automatically, reaching a steady state. As the ribosome is continuously synthesized, and reaches the terminus of the mRNA, it meets the "unanimous codon", whereby the mRNA is then reproduced. The synthesized protein is finally released, as shown in Figure 3.

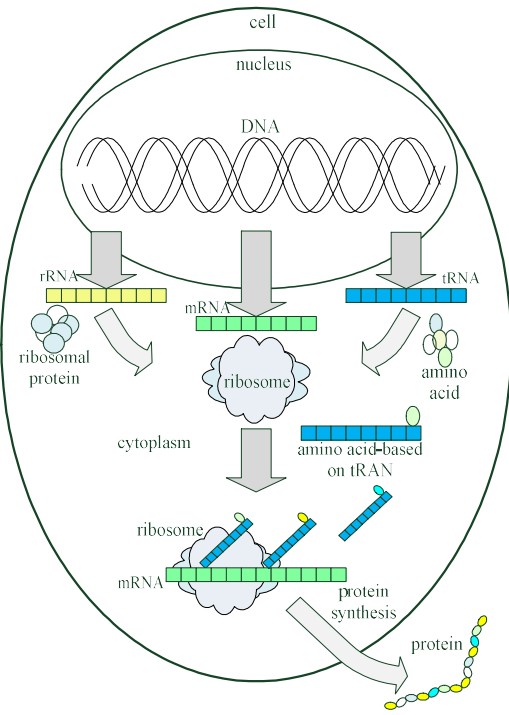

**Figure 3.** Schematic diagram of protein synthesis.

The future research on life aims at what proteins there are, how they function, and how they interact with environment stimulation. The proteome is also known as proteinoplast, i.e., the combinations of all proteins produced by human cells and tissues. While the genome is the composition of the complete genetic information of organisms, it merely carries the formula for producing proteins; thus, protein cells are composed of footstones, which take charge of most of the work. In addition, the differences among heterogeneous cells result from proteins. While all cells of a person have almost the same genome, each kind of cell has different proteins produced by activating genes. In the same way, ill cells often produce some proteins that healthy cells do not have, and vice versa.

*3.2. The Control Strategy of Refrigerating Chambers*

Table 1 shows the control strategy of refrigerating chambers. There are two strategies for optimizing the energy saving of refrigerating chambers; one is to operate the refrigerant circulation of the refrigerating machine set at the optimum condition; the other one is to operate the compressor of the refrigerating machine set at approximately the status of full load. The optimum condition for refrigerant circulation represents the maximum refrigerating capacity (system heat absorption capacity, kW) and minimum power consumed by the compressor (kW) in the refrigeration cycle, as shown in Figure 1, where the ratio of the two values is the *COP*. To maximize the refrigerating capacity of the refrigeration cycle, the high pressure should be as low as possible, but the low pressure should not drop excessively. To minimize the power consumed by the compressor, the low pressure should be as high as possible; however, as the evaporating temperature must meet the refrigerating chamber temperature, it should not be too high. On the other hand, for basic mechanical loss, if the compressor runs in the low-load condition, the overall efficiency will decrease.

**Table 1.** The control strategy of refrigerating chambers.

| Item | Strategic Objective | Description |
|---|---|---|
| 1 | Optimum refrigerant recirculating status | (1) The electromagnetic valve of the refrigerating chamber is turned off before the upper or lower temperature limit is exceeded. As the load of the refrigeration cycle system decreases, the power input will decrease. |
| | | (2) As the refrigerant circulation decreases, the load of the compressor decreases, such that the power input work decreases, and the high/low pressure of the system is influenced. The expansion valve opening is increased, while the refrigerant flow decreases; thus, as the high pressure drops and the low pressure remains, the power input of the compressor decreases. |
| 2 | Full load of compressor | (1) When the status of the reorganization valve in the refrigerating chamber is on, the refrigerant enters the refrigerating chamber via the expansion valve to absorb the heat energy in the chamber, while the overheated refrigerant, which absorbed the heat energy, returns to the low-pressure end of the compressor. Therefore, the more electromagnetic valves are turned on, the more overheated refrigerant returns to the low-pressure suction end of the compressor, and the refrigeration load of the refrigeration cycle system increases. The electromagnetic valve of the refrigerating chamber is turned on before the upper or lower temperature limit is exceeded, and the running compressor is in the full-load status. |
| | | (2) On the other hand, the electromagnetic valve of a refrigerating chamber is turned off before the upper or lower temperature limit is exceeded, and the total load of the refrigeration cycle system is reduced, which switches off the compressor under low load, while the running compressor remains in full-load conditions at any time. |

In order to implement the aforesaid two energy-saving control strategies, this study uses BPA to change the compressor and the on/off status of the electromagnetic valve to change the refrigerating capacity and chamber load. In other words, the status of the compressor and electromagnetic valve are changed by the controller before the refrigerating chamber temperature exceeds the upper or lower temperature limit, thereby regulating the refrigerating capacity and refrigeration load of the system, allowing the refrigeration system to work at high *COP* for a long time. However, the system cycling loss and machine member abrasion loss must be considered when turning the electromagnetic valve on/off; therefore, the electromagnetic valve should not be switched too frequently, meaning that a criterion must be established to evaluate the benefit of switching the electromagnetic valve status by the controller, in order to attain the goal of energy saving.

The objective function is defined as the operating efficiency of the refrigerating chambers, as shown in Equation (1).

$$COP^{t_n} = \frac{\sum\limits_{t=1}^{t_n} \sum\limits_{i=1}^{N_e} \left[ V_i^t Q_i^t - V_i^t (1 - V_i^{t-1}) S_{qi}^t \right]}{\sum\limits_{i=1}^{t_n} \sum\limits_{j=1}^{N_c} \left[ U_j^t P_j^t + U_j^t (1 - U_j^{t-1}) S_{pj}^t \right]} \tag{1}$$

where $COP^{t_n}$ is the average $COP$ in $t_n$, $t_n$ is the total number of time stages, $t$ is the time stage of status, $i$ is the $i$-th refrigerating chamber, $j$ is the $j$-th compressor, $N_e$ is the number of refrigerating chambers, $N_c$ is the number of compressors, $Q_i^t$ is the refrigerating capacity of the $i$-th refrigerating chamber at time $t$, $V_i^t$ is the status of the electromagnetic valve (0: off, 1: on), $U_j^t$ is the compressor operating status (0: off, 1: on), $S_{qi}^t$ is the switching loss when the status of the electromagnetic valve is switched from 0 to 1, $S_{pj}^t$ is the switching loss when the compressor status is switched from 0 to 1 and $P_j^t$ is the power consumed by the compressor within time $t$.

The constraints are described by the following conditions:

1.  Temperature limitation of refrigerating chamber,

$$T_{li} \le T_i^t \le T_{ui}, \tag{2}$$

where $T_i^t$ is the temperature of refrigerating chambers, $T_{li}$ is the preset floor temperature of refrigerating chambers, and $T_{ui}$ is the preset ceiling temperature of refrigerating chambers;

2.　Shortest turn-off time limitation of electromagnetic valve.

In the course of the cold supply for refrigerating chambers, as the evaporator surface temperature is lower than the condensation temperature of moisture, the air moisture condenses on the evaporator and frosts, and the humidity decreases. When the electromagnetic valve is turned off, it can be turned on again in at least seven minutes, so that the humidifier humidifies the indoor air. The cumulative turn-off time $I_{ei}^t$ of the refrigerating chamber's electromagnetic valve is calculated using Equations (3) and (4).

$$I_{ei}^t = 0, \ V_i^t = 1; \tag{3}$$

$$I_{ei}^t = I_{ei}^t + 1, \quad V_i^t = 0. \tag{4}$$

This study regards the optimal operating strategy of a refrigeration system as the solution of the objective function, i.e., the proteome. One single protein contains a single gene combination, representing the on/off combination of an electromagnetic valve in a time stage. The protein formation process generates different proteomes, and the preferential mechanism selects the proteome with a high health index, i.e., the optimal solution of the objective function. The BPA performs binary coding of different electromagnetic valve on/off statuses under different loads of the refrigerating chambers. The obtained feasible solutions must be checked by the cooling capacity ($Q_c$), refrigeration capacity ($Q_e$), and power consumption ($P_c$) identification, as expressed by Equation (5).

$$Q_c \geq Q_e + 860P_c \tag{5}$$

Only the solution satisfying the mass–energy conservation is the feasible solution. Afterward, all feasible solutions are ordered, and the superior solution is selected as the first evolved solution, which is iterated until the algorithm converges.

### 3.3. Implementation of BPA

The procedure of BPA applied in the energy saving of refrigerating cambers is below.

1.　Gene transcription—mRAN produced

By considering the 24 intervals, the initial mRAN is obtained by assigning a binary digit for the status of refrigerant electromagnetic valve of each chamber as shown in Figure 4, whereby "1" and "0" are the status of the refrigerant electromagnetic valve to be on and off, respectively.

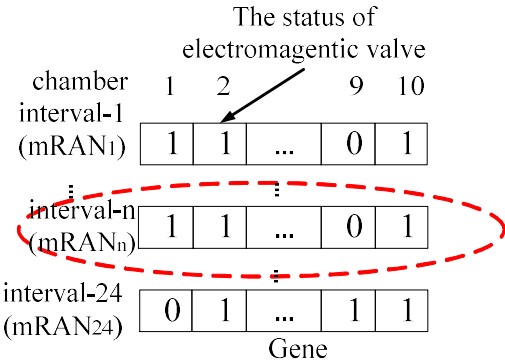

**Figure 4.** mRAN produced.

2. Gene translation—mRAN translated

The chambers check the status of the electromagnetic valve in sequence. If the status is "0", do nothing. If the status is "1", transfer the status in random. When Rand(1,0) > 0.5, transfer the status from 1 to 0 as shown in Figure 5.

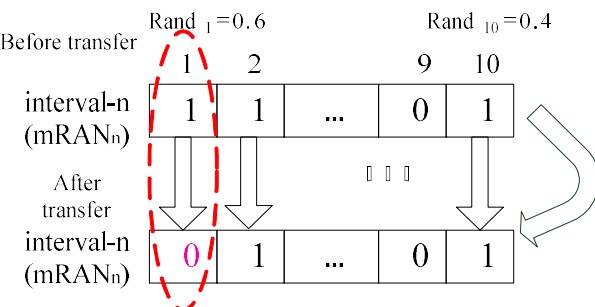

**Figure 5.** mRAN translated.

3. Amino acid match

Amino acids are brought up by tRNA with mRNA to complete the translation. Each transferred status of the electromagnetic valves must be checked using Equations (2)–(5) to determine whether they violated any constraints. If the transferred status met all constraints, it is confirmed that the status can be transferred, as shown in Figure 6.

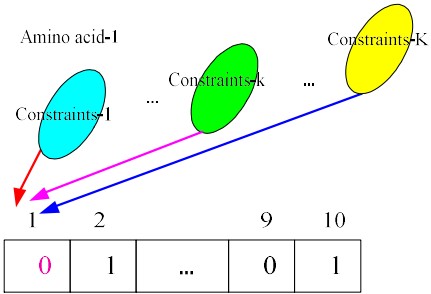

**Figure 6.** Amino acid match.

4. Protein synthesis—the amino acid is synthesized into a protein.

After Step 3, the refrigerating chamber which is in the on status will have a load. The load is from the goods stored in the refrigerating chamber. If the refrigerating chamber with the electromagnetic valve closed has no load, the status of the electromagnetic valve is changed to 0, as shown in Figure 7.

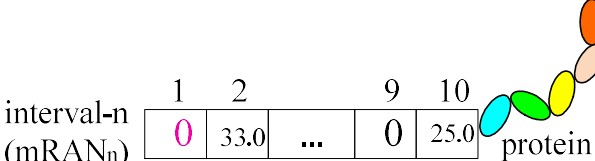

**Figure 7.** Protein synthesis.

5. Calculate *COP*

Use Equation (1) to calculate the *COP* value for all refrigerating chambers in the single interval. In other words, put each load value of the refrigerating chambers into Equation (1) in the single interval. Thus, each protein will represent a single $COP^i$ ($i = 1 \cdots t_n$) value.

6.  Picking good protein

In order to get the best operation strategy of refrigerating chambers in the same time interval, repeat Steps 2 to 5 until a good solution is picked. Each protein represents a solution in a time interval.

7.  Picking the best proteome

Substituting each of the optimum solutions of interval 1 to interval 24 into Equation (5) calculates the optimum operating *COP* value throughout the day, as shown in Figure 8.

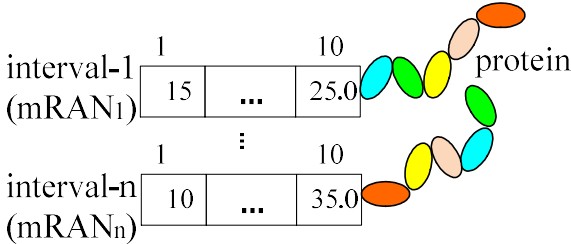

**Figure 8.** Picking the best proteome.

Figure 9 illustrates the procedure of the BPA.

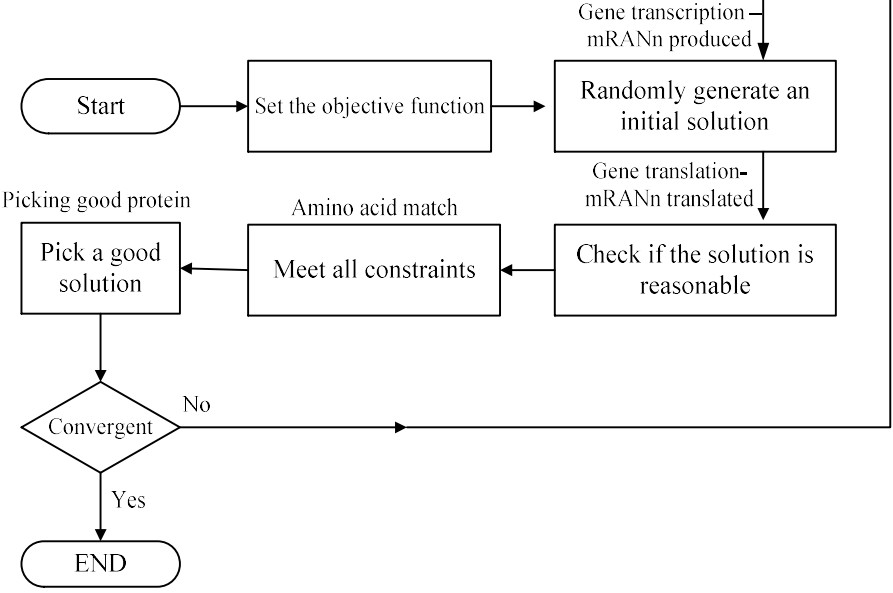

**Figure 9.** The procedure of the binary proteome algorithm (BPA).

## 4. Case Study

The proposed algorithm was tested on refrigerating chambers of the food company, where two reciprocating compressors of the same capacity 50 RT (176 kW) were connected in parallel; the total refrigerating capacity was 100 RT (352 kW), there were 10 refrigerating chambers, and the capacity of each refrigerating chamber was 10 RT (35.2 kW). This test performed the BPA to optimize the energy control strategy according to the loads and refrigerating capacity of the compressors of various refrigerating chambers at each time stage, as shown in Table 2. This paper takes the time stage from 11:00 a.m. to 4:00 p.m. of the refrigerating chamber in a day as the simulation test example. The five hours are subdivided into small time stages of five minutes, in order to test the real-time efficiency. Table 2 shows the load change from 11:00 a.m. to 1:00 p.m., while Table 3 shows the on/off status of the compressor of this system under classical control. The program of BPA was written in a MATLAB

working environment. The tests were carried out on an Intel Core i5-3470s 2.9-GHz central processing unit (CPU) with 8 GB of dynamic random-access memory (DRAM).

**Table 2.** The load change of refrigerating chamber (kW).

| Time Stage | Chamber / Time | 1 | 2 | 3 | 4 | 5 | 6 | 7 | 8 | 9 | 10 | Total Load |
|---|---|---|---|---|---|---|---|---|---|---|---|---|
| 1 | 11:00–11:05 a.m. | 30.0 | 33.0 | 34.5 | 35.0 | 34.0 | 0.0 | 0.0 | 0.0 | 0.0 | 0.0 | 166.5 |
| 2 | 11:05–11:10 a.m. | 30.0 | 33.0 | 34.5 | 35.0 | 34.0 | 0.0 | 0.0 | 0.0 | 0.0 | 0.0 | 166.5 |
| 3 | 11:10–11:15 a.m. | 30.0 | 33.0 | 34.5 | 35.0 | 34.0 | 0.0 | 0.0 | 0.0 | 0.0 | 0.0 | 166.5 |
| 4 | 11:15–11:20 a.m. | 25.0 | 34.0 | 34.5 | 35.0 | 34.0 | 0.0 | 0.0 | 0.0 | 0.0 | 0.0 | 162.5 |
| 5 | 11:20–11:25 a.m. | 25.0 | 34.0 | 30.0 | 35.0 | 34.0 | 0.0 | 0.0 | 10.0 | 25.0 | 0.0 | 193.0 |
| 6 | 11:25–11:30 a.m. | 32.0 | 34.0 | 30.0 | 35.0 | 34.0 | 0.0 | 0.0 | 10.0 | 25.0 | 0.0 | 200.0 |
| 7 | 11:30–11:35 a.m. | 33.0 | 35.0 | 30.0 | 35.0 | 34.0 | 0.0 | 0.0 | 5.0 | 25.0 | 0.0 | 197.0 |
| 8 | 11:35–11:40 a.m. | 33.0 | 35.0 | 34.5 | 35.0 | 34.0 | 32.0 | 33.0 | 24.0 | 25.0 | 0.0 | 285.5 |
| 9 | 11:40–11:45 a.m. | 33.0 | 34.0 | 34.5 | 35.0 | 34.0 | 32.0 | 33.0 | 24.0 | 25.0 | 0.0 | 284.5 |
| 10 | 11:45–11:50 a.m. | 34.0 | 34.0 | 34.5 | 35.0 | 34.0 | 32.0 | 33.0 | 24.0 | 25.0 | 15.0 | 300.5 |
| 11 | 11:50–11:55 a.m. | 34.0 | 26.0 | 34.5 | 35.0 | 34.0 | 32.0 | 33.0 | 24.0 | 25.0 | 15.0 | 292.5 |
| 12 | 11:55 a.m.–12:00 p.m. | 34.0 | 24.0 | 34.5 | 27.0 | 29.0 | 32.0 | 33.0 | 24.0 | 27.0 | 15.0 | 279.5 |
| 13 | 12:00–12:05 p.m. | 34.0 | 24.0 | 25.0 | 27.0 | 29.0 | 32.0 | 33.0 | 24.0 | 27.0 | 15.0 | 270.0 |
| 14 | 12:05–12:10 p.m. | 34.0 | 24.0 | 25.0 | 27.0 | 29.0 | 32.0 | 33.0 | 24.0 | 27.0 | 15.0 | 270.0 |
| 15 | 12:10–12:15 p.m. | 35.0 | 24.0 | 25.0 | 27.0 | 29.0 | 32.0 | 33.0 | 24.0 | 27.0 | 15.0 | 271.0 |
| 16 | 12:15–12:20 p.m. | 35.0 | 15.0 | 25.0 | 27.0 | 29.0 | 32.0 | 19.0 | 24.0 | 27.0 | 15.0 | 248.0 |
| 17 | 12:20–12:25 p.m. | 25.0 | 15.0 | 25.0 | 27.0 | 29.0 | 32.0 | 19.0 | 24.0 | 27.0 | 15.0 | 238.0 |
| 18 | 12:25–12:30 p.m. | 23.0 | 15.0 | 25.0 | 14.0 | 29.0 | 32.0 | 19.0 | 24.0 | 27.0 | 15.0 | 223.0 |
| 19 | 12:30–12:35 p.m. | 22.0 | 15.0 | 25.0 | 14.0 | 29.0 | 32.0 | 19.0 | 24.0 | 27.0 | 15.0 | 222.0 |
| 20 | 12:35–12:40 p.m. | 21.0 | 15.0 | 0.0 | 14.0 | 29.0 | 23.0 | 19.0 | 31.0 | 27.0 | 15.0 | 194.0 |
| 21 | 12:40–12:45 p.m. | 10.0 | 22.0 | 0.0 | 14.0 | 17.0 | 23.0 | 19.0 | 31.0 | 0.0 | 15.0 | 151.0 |
| 22 | 12:45–12:50 p.m. | 10.0 | 22.0 | 0.0 | 14.0 | 17.0 | 23.0 | 19.0 | 31.0 | 0.0 | 15.0 | 151.0 |
| 23 | 12:50–12:55 p.m. | 10.0 | 22.0 | 0.0 | 14.0 | 17.0 | 23.0 | 19.0 | 31.0 | 0.0 | 25.0 | 161.0 |
| 24 | 12:55–1:00 p.m. | 10.0 | 22.0 | 0.0 | 14.0 | 17.0 | 23.0 | 19.0 | 31.0 | 0.0 | 25.0 | 161.0 |

**Table 3.** Compressor on/off status under classical control of refrigerating chamber. BPAbinary proteome algorithm.

| Time Stage | Classical Control of #1 and #2 Compressor On/Off Status | | BPA Control of #1 and #2 Compressor On/Off Status | |
|---|---|---|---|---|
| | #1 Compressor Status | #2 Compressor Status | #1 Compressor Status | #2 Compressor Status |
| 1 | On | Off | On | Off |
| 2 | On | Off | On | Off |
| 3 | On | Off | On | Off |
| 4 | On | Off | On | Off |
| 5 | On | On | On | Off |
| 6 | On | On | On | Off |
| 7 | On | On | On | Off |
| 8 | On | On | On | On |
| 9 | On | On | On | On |
| 10 | On | On | On | On |
| 11 | On | On | On | On |
| 12 | On | On | On | On |
| 13 | On | On | On | On |
| 14 | On | On | On | On |
| 15 | On | On | On | On |
| 16 | On | On | On | On |
| 17 | On | On | On | On |
| 18 | On | On | On | On |
| 19 | On | On | On | On |
| 20 | On | On | On | Off |
| 21 | On | Off | On | Off |
| 22 | On | Off | On | Off |
| 23 | On | Off | On | Off |
| 24 | On | Off | On | Off |

### 4.1. The Control Strategy Analysis of the Refrigerating Chamber

As shown in Table 3, the on/off status of the compressor obtained by classical control on/off is shown on the left, while the compressor on/off status obtained by BPA is shown on the right. According to the data in Table 3, the #2 compressor was off at time stages 5, 6, 7, and 20. The power consumption of the refrigerating chamber system decreased, and efficiency increased across the time stages. The #2 compressor was off at time stages 5, 6, 7, and 20, as the refrigerant electromagnetic valve of some refrigerating chambers met the off condition in these time stages according to the BPA. Therefore, these refrigerating chambers, where the electromagnetic valve was turned off, had no load on the system in these time stages. As the refrigerant did not flow into the refrigerating chamber to absorb heat energy, the refrigerated chamber was free of load, as shown in Table 4.

**Table 4.** The load changes of refrigerating chamber after the energy control strategy(kW).

| Time Stage | Chamber / Time | 1 | 2 | 3 | 4 | 5 | 6 | 7 | 8 | 9 | 10 | Total Load |
|---|---|---|---|---|---|---|---|---|---|---|---|---|
| 1 | 11:00–11:05 a.m. | 30.0 | 33.0 | 34.5 | 35.0 | 34.0 | 0.0 | 0.0 | 0.0 | 0.0 | 0.0 | 166.5 |
| 2 | 11:05–11:10 a.m. | 30.0 | 33.0 | 34.5 | 35.0 | 34.0 | 0.0 | 0.0 | 0.0 | 0.0 | 0.0 | 166.5 |
| 3 | 11:10–11:15 a.m. | 30.0 | 33.0 | 34.5 | 35.0 | 34.0 | 0.0 | 0.0 | 0.0 | 0.0 | 0.0 | 166.5 |
| 4 | 11:15–11:20 a.m. | 25.0 | 34.0 | 34.5 | 35.0 | 34.0 | 0.0 | 0.0 | 0.0 | 0.0 | 0.0 | 162.5 |
| 5 | 11:20–11:25 a.m. | 25.0 | 34.0 | 30.0 | 35.0 | 34.0 | 0.0 | 0.0 | 0.0 | 0.0 | 0.0 | 168.0 |
| 6 | 11:25–11:30 a.m. | 32.0 | 34.0 | 30.0 | 35.0 | 34.0 | 0.0 | 0.0 | 0.0 | 0.0 | 0.0 | 175.0 |
| 7 | 11:30–11:35 a.m. | 33.0 | 35.0 | 30.0 | 35.0 | 34.0 | 0.0 | 0.0 | 0.0 | 0.0 | 0.0 | 172.0 |
| 8 | 11:35–11:40 a.m. | 33.0 | 35.0 | 34.5 | 35.0 | 34.0 | 32.0 | 33.0 | 24.0 | 25.0 | 0.0 | 285.5 |
| 9 | 11:40–11:45 a.m. | 33.0 | 34.0 | 34.5 | 35.0 | 34.0 | 32.0 | 33.0 | 24.0 | 25.0 | 0.0 | 284.5 |
| 10 | 11:45–11:50 a.m. | 34.0 | 34.0 | 34.5 | 35.0 | 34.0 | 32.0 | 33.0 | 24.0 | 25.0 | 15.0 | 300.5 |
| 11 | 11:50–11:55 a.m. | 34.0 | 26.0 | 34.5 | 35.0 | 34.0 | 32.0 | 33.0 | 24.0 | 25.0 | 15.0 | 292.5 |
| 12 | 11:55 a.m.–12:00 p.m. | 34.0 | 24.0 | 34.5 | 27.0 | 29.0 | 32.0 | 33.0 | 24.0 | 27.0 | 15.0 | 279.5 |
| 13 | 12:00–12:05 p.m. | 34.0 | 24.0 | 25.0 | 27.0 | 29.0 | 32.0 | 33.0 | 24.0 | 27.0 | 15.0 | 270.0 |
| 14 | 12:05–12:10 p.m. | 34.0 | 24.0 | 25.0 | 27.0 | 29.0 | 32.0 | 33.0 | 24.0 | 27.0 | 15.0 | 270.0 |
| 15 | 12:10–12:15 p.m. | 35.0 | 24.0 | 25.0 | 27.0 | 29.0 | 32.0 | 33.0 | 24.0 | 27.0 | 15.0 | 271.0 |
| 16 | 12:15–12:20 p.m. | 35.0 | 15.0 | 25.0 | 27.0 | 29.0 | 32.0 | 19.0 | 24.0 | 27.0 | 15.0 | 248.0 |
| 17 | 12:20–12:25 p.m. | 25.0 | 15.0 | 25.0 | 27.0 | 29.0 | 32.0 | 19.0 | 24.0 | 27.0 | 15.0 | 238.0 |
| 18 | 12:25–12:30 p.m. | 23.0 | 15.0 | 25.0 | 14.0 | 29.0 | 32.0 | 19.0 | 24.0 | 27.0 | 15.0 | 223.0 |
| 19 | 12:30–12:35 p.m. | 22.0 | 15.0 | 25.0 | 14.0 | 29.0 | 32.0 | 19.0 | 24.0 | 27.0 | 15.0 | 222.0 |
| 20 | 12:35–12:40 p.m. | 21.0 | 15.0 | 0.0 | 14.0 | 29.0 | 23.0 | 19.0 | 31.0 | 0.0 | 0.0 | 152.0 |
| 21 | 12:40–12:45 p.m. | 10.0 | 22.0 | 0.0 | 14.0 | 17.0 | 23.0 | 19.0 | 31.0 | 0.0 | 15.0 | 151.0 |
| 22 | 12:45–12:50 p.m. | 10.0 | 22.0 | 0.0 | 14.0 | 17.0 | 23.0 | 19.0 | 31.0 | 0.0 | 15.0 | 151.0 |
| 23 | 12:50–12:55 p.m. | 10.0 | 22.0 | 0.0 | 14.0 | 17.0 | 23.0 | 19.0 | 31.0 | 0.0 | 25.0 | 161.0 |
| 24 | 12:55–1:00 p.m. | 10.0 | 22.0 | 0.0 | 14.0 | 17.0 | 23.0 | 19.0 | 31.0 | 0.0 | 25.0 | 161.0 |

According to Table 4, the status of the electromagnetic valve was off when the No. 9 refrigerating chamber was in time stages 5, 6, 7, and 20, and the electromagnetic valve was also off when the No. 10 refrigerating chamber was in time stage 20. Therefore, the total system load decreased to 168 kW in time stage 5, the total system load decreased to 175 kW in time stage 6, the total system load decreased to 172 kW in time stage 7, and the total system load decreased to 152 kW in time stage 20. As the total load decreased in the aforesaid four time stages, one compressor was enough to supply the total load for the system; thus, the #2 compressor was switched off, the total power consumption of the system decreased, and the efficiency increased.

### 4.2. The Energy-Saving Analysis of the Refrigerating Chamber

Table 5 shows that the temperatures in the refrigerating chamber corresponded to the changes in the refrigerating chamber load (kW) in Table 2. As the upper temperature limit of the refrigerating chamber was −10 °C, and the lower limit was −20 °C, the energy-saving strategy of this study was to turn it off below −15 °C, in order to avoid the goods in the chamber deteriorating. Table 6 shows the comparison among system load, power consumption, and *COP*.

**Table 5.** Temperature changes of refrigerating chamber load in various time stages (°C).

| Time Stage | Chamber / Time | 1 | 2 | 3 | 4 | 5 | 6 | 7 | 8 | 9 | 10 |
|---|---|---|---|---|---|---|---|---|---|---|---|
| 1 | 11:00–11:05 a.m. | −11.5 | −11.0 | −11.0 | −10.5 | −10.5 | −10.0 | −10.0 | −10.0 | −10.0 | −10.0 |
| 2 | 11:05–11:10 a.m. | −11.5 | −11.0 | −11.0 | −10.5 | −10.5 | −10.0 | −10.0 | −10.0 | −10.0 | −10.0 |
| 3 | 11:10–11:15 a.m. | −11.5 | −11.0 | −11.0 | −10.5 | −10.5 | −10.0 | −10.0 | −10.0 | −10.0 | −10.0 |
| 4 | 11:15–11:20 a.m. | −11.5 | −11.0 | −11.0 | −10.5 | −10.5 | −10.0 | −10.0 | −10.0 | −10.0 | −10.0 |
| 5 | 11:20–11:25 a.m. | −12.0 | −11.0 | −11.0 | −10.5 | −10.5 | −10.0 | −10.0 | −10.0 | −15.5 | −10.0 |
| 6 | 11:25–11:30 a.m. | −12.0 | −11.0 | −11.0 | −10.5 | −10.5 | −10.0 | −10.0 | −11.0 | −15.0 | −10.0 |
| 7 | 11:30–11:35 a.m. | −12.0 | −11.0 | −11.0 | −10.5 | −10.5 | −10.0 | −10.5 | −11.0 | −15.0 | −10.0 |
| 8 | 11:35–11:40 a.m. | −12.0 | −11.0 | −11.0 | −10.5 | −10.5 | −10.5 | −10.5 | −11.0 | −14.5 | −10.0 |
| 9 | 11:40–11:45 a.m. | −12.0 | −11.0 | −11.0 | −10.5 | −10.5 | −10.5 | −10.5 | −11.0 | −14.5 | −10.0 |
| 10 | 11:45–11:50 a.m. | −12.0 | −12.5 | −11.0 | −10.5 | −10.5 | −10.5 | −10.5 | −11.0 | −14.5 | −12.0 |
| 11 | 11:50–11:55 a.m. | −12.0 | −12.5 | −11.0 | −10.5 | −10.5 | −10.5 | −10.5 | −13.0 | −14.5 | −12.0 |
| 12 | 11:55 a.m.–12:00 p.m. | −12.0 | −12.5 | −11.0 | −10.5 | −10.5 | −10.5 | −10.5 | −12.0 | −14.5 | −12.0 |
| 13 | 12:00–12:05 p.m. | −14.0 | −12.5 | −11.0 | −13.5 | −10.5 | −10.5 | −10.5 | −12.0 | −14.5 | −12.0 |
| 14 | 12:05–12:10 p.m. | −14.0 | −12.5 | −13.0 | −13.5 | −11.0 | −10.5 | −10.5 | −12.0 | −13.0 | −13.0 |
| 15 | 12:10–12:15 p.m. | −14.0 | −12.5 | −13.0 | −13.5 | −11.0 | −10.5 | −10.5 | −12.0 | −13.0 | −13.0 |
| 16 | 12:15–12:20 p.m. | −14.0 | −13.0 | −13.0 | −13.5 | −11.0 | −10.5 | −10.5 | −12.0 | −13.0 | −14.0 |
| 17 | 12:20–12:25 p.m. | −14.0 | −13.0 | −13.0 | −13.5 | −11.0 | −10.5 | −10.5 | −12.0 | −13.0 | −14.0 |
| 18 | 12:25–12:30 p.m. | −14.0 | −13.0 | −13.0 | −13.5 | −11.0 | −10.5 | −12.0 | −12.0 | −13.0 | −14.5 |
| 19 | 12:30–12:35 p.m. | −13.0 | −13.0 | −13.0 | −13.5 | −11.0 | −10.5 | −12.0 | −12.0 | −14.0 | −15.0 |
| 20 | 12:35–12:40 p.m. | −13.0 | −13.0 | −13.0 | −13.5 | −11.0 | −10.5 | −12.0 | −12.0 | −15.0 | −15.7 |
| 21 | 12:40–12:45 p.m. | −13.0 | −13.0 | −13.0 | −14.0 | −12.0 | −11.5 | −12.0 | −11.5 | −15.0 | −14.5 |
| 22 | 12:45–12:50 p.m. | −13.0 | −11.0 | −14.0 | −14.0 | −12.0 | −11.5 | −12.0 | −11.5 | −10.0 | −14.5 |
| 23 | 12:50–12:55 p.m. | −13.0 | −11.0 | −14.0 | −14.0 | −12.0 | −11.5 | −12.0 | −11.5 | −10.0 | −14.5 |
| 24 | 12:55–1:00 p.m. | −13.0 | −11.0 | −14.0 | −14.0 | −12.0 | −11.5 | −12.0 | −11.5 | −10.0 | −14.5 |

**Table 6.** The comparison among system load, power consumption, and coefficient of performance (*COP*).

| Time Stage | Time | Before Implementation of Energy Control Strategy | | | After Implementation of Energy Control Strategy | | |
|---|---|---|---|---|---|---|---|
| | | System Load | Power Consumption | *COP* | System Load | Power Consumption | *COP* |
| 1 | 11:00–11:05 a.m. | 166.5 | 75.7 | 2.2 | 166.5 | 75.7 | 2.2 |
| 2 | 11:05–11:10 a.m. | 166.5 | 75.7 | 2.2 | 166.5 | 75.7 | 2.2 |
| 3 | 11:10–11:15 a.m. | 166.5 | 75.7 | 2.2 | 166.5 | 75.7 | 2.2 |
| 4 | 11:15–11:20 a.m. | 162.5 | 75.7 | 2.2 | 162.5 | 75.7 | 2.2 |
| 5 | 11:20–11:25 a.m. | 183.0 | 114.4 | 1.6 | 168.0 | 70.0 | 2.4 |
| 6 | 11:25–11:30 a.m. | 190.0 | 111.8 | 1.7 | 175.0 | 76.1 | 2.3 |
| 7 | 11:30–11:35 a.m. | 192.0 | 112.9 | 1.7 | 172.0 | 74.8 | 2.3 |
| 8 | 11:35–11:40 a.m. | 285.5 | 150.3 | 1.8 | 285.5 | 150.3 | 1.8 |
| 9 | 11:40–11:45 a.m. | 284.5 | 149.7 | 1.8 | 284.5 | 149.7 | 1.8 |
| 10 | 11:45–11:50 a.m. | 300.5 | 158.2 | 1.9 | 300.5 | 158.2 | 1.9 |
| 11 | 11:50–11:55 a.m. | 292.5 | 154.0 | 1.9 | 292.5 | 154.0 | 1.9 |
| 12 | 11:55 a.m.–12:00 p.m. | 279.5 | 155.3 | 1.8 | 279.5 | 155.3 | 1.8 |
| 13 | 12:00–12:05 p.m. | 270.0 | 150.0 | 1.8 | 270.0 | 150.0 | 1.8 |
| 14 | 12:05–12:10 p.m. | 270.0 | 150.0 | 1.8 | 270.0 | 150.0 | 1.8 |
| 15 | 12:10–12:15 p.m. | 271.0 | 150.5 | 1.8 | 271.0 | 150.5 | 1.8 |
| 16 | 12:15–12:20 p.m. | 248.0 | 137.8 | 1.8 | 248.0 | 137.8 | 1.8 |
| 17 | 12:20–12:25 p.m. | 238.0 | 132.2 | 1.8 | 238.0 | 132.2 | 1.8 |
| 18 | 12:25–12:30 p.m. | 223.0 | 124.0 | 1.8 | 223.0 | 124.0 | 1.8 |
| 19 | 12:30–12:35 p.m. | 222.0 | 123.3 | 1.8 | 222.0 | 123.3 | 1.8 |
| 20 | 12:35–12:40 p.m. | 194.0 | 121.2 | 1.6 | 152.0 | 69.1 | 2.2 |
| 21 | 12:40–12:45 p.m. | 151.0 | 68.6 | 2.2 | 151.0 | 68.6 | 2.2 |
| 22 | 12:45–12:50 p.m. | 151.0 | 68.6 | 2.2 | 151.0 | 68.6 | 2.2 |
| 23 | 12:50–12:55 p.m. | 161.0 | 73.2 | 2.2 | 161.0 | 73.2 | 2.2 |
| 24 | 12:55–1:00 p.m. | 161.0 | 73.2 | 2.2 | 161.0 | 73.2 | 2.2 |
| | Average value | - | 115.92 | 1.92 | - | 108.82 | 2.03 |

The summary of the energy-saving analysis is described below.

1. According to the comparison between Tables 2 and 5, there are no goods stored in the No. 6–10 refrigerating chambers in the first few time stages of the test example, and the maximum temperature limit of −10 °C was maintained.

2.  According to the comparison among Tables 2, 4 and 5, when the No. 9 refrigerating chamber was in time stages 5, 6, 7, and 20, and the No. 10 refrigerating chamber was in time stage 20, the status of the electromagnetic valve was changed from on to off by the energy control strategy, as the temperature met the limitation set in the time stages.

3.  According to Tables 2–4, the refrigerant control electromagnetic valve was changed from on to off after the energy control strategy was performed in time stages 5, 6, and 7; thus, the total load of the system decreased, meaning the total power consumption decreased.

4.  According to the comparison between Tables 3 and 6, the total load was 183 kW in time stage 5, the total power consumption of the system was 114.4 kW, and the *COP* value was 1.6. After the energy conservation strategy was performed in the same time stage, the total load decreased to 168 kW, the total power consumption of the system was 70 kW, and the *COP* value was 2.4.

5.  According to Tables 2–4, as the electromagnetic valve of the No. 9 refrigerating chamber was turned off in time stage 5, the #2 compressor was switched off by the control strategy, according to the capacity regulation determination, as the load decreased.

6.  When the No. 9 refrigerating chamber was in time stages 6, 7, and 20, and the No. 10 refrigerating chamber was in time stage 20, the electromagnetic valve was turned off by the control strategy for the same reason, and the #2 compressor was shut down. This action not only increased the *COP* value in these stages, it also reduced the total power consumption of the system.

7.  As shown in Figure 10, in time stages 5, 6, 7, and 20, as the temperatures of the No. 9 and No. 10 refrigerating chambers met the electromagnetic valve off condition, the electromagnetic valve was turned off, and the refrigerating chamber load decreased; thus, the total system load decreased.

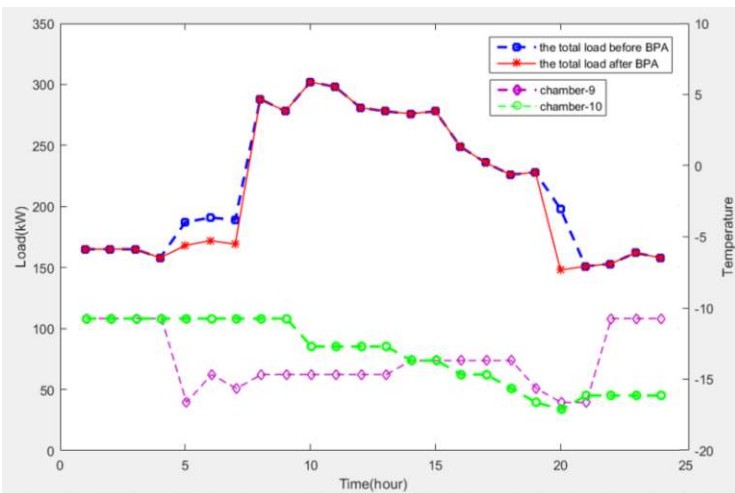

**Figure 10.** Temperature and load changes of the No. 9 and No. 10 refrigerating chambers.

### 4.3. BPA Performance Test

The test system used in this paper was expressed as Equation (1), and the operating limitations of the refrigeration system of various refrigerating chambers were expressed as Equations (2)–(4); then, the BPA, genetic algorithm (GA), and simulated annealing (SA) were simulated in the same environment. This case used BPA to compute five different total numbers of time stages, whereby each time stage was tested 50 times by simulation, and each test had 50 iterations. Table 7 shows the simulation test results of the five time stages. Figure 11 shows the convergent characteristics of the different time stages.

**Table 7.** BPA test for different numbers of time stages.

| Item | Number of Time Stages | Time Stage 12 | Time Stage 24 | Time Stage 36 | Time Stage 48 | Time Stage 60 |
|---|---|---|---|---|---|---|
| Optimum average *COP* | | 2.1 | 2.0 | 2.2 | 2.3 | 2.4 |
| Worst average *COP* | | 1.8 | 1.8 | 1.9 | 2.0 | 2.0 |
| Times reaching optimal value | | 15 | 17 | 21 | 27 | 22 |
| Average number of iterations for occurrence of optimal value | | 33 | 32 | 35 | 28 | 25 |
| Average execution time (s) | | 24.5 | 74.5 | 208.0 | 297.7 | 387.7 |

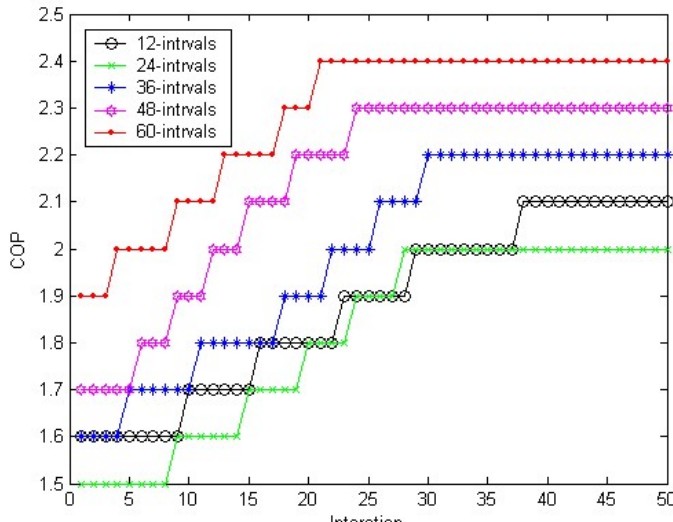

**Figure 11.** The convergent characteristics of the different time stages.

This case used the time stage 60 example as the comparison basis. BPA solved the energy-saving optimization problem, and the result was compared with the results of GA and SA, as shown in Figure 12. According to the convergence curves of the various algorithms, the BPA had a better convergence rate and a more optimal *COP* than the other two algorithms.

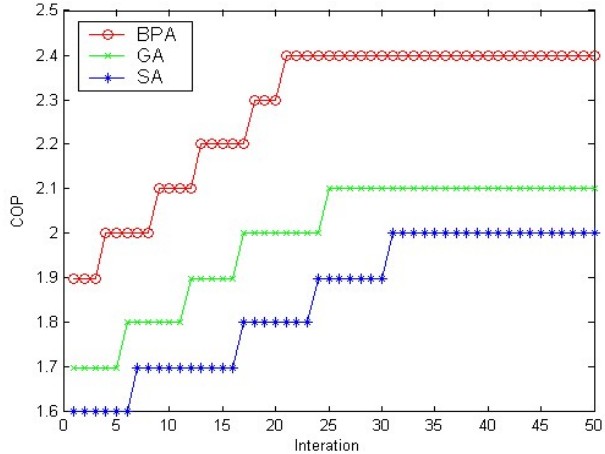

**Figure 12.** BPA performance test.

*4.4. Test for BPA Considering Switching Loss*

This case used the time stage 48 (11:00 a.m.–3:00 p.m.) example as the comparison basis, where switching loss was adjusted to 1.2 times that of the original, as shown in Table 8. When the switching loss was increased, the test result showed that the optimal *COP* value decreased, the worst *COP* value

decreased, the optimal value was easier to be obtained, the convergence time was shorter, and there was a slight difference in search time.

**Table 8.** Switching loss test.

| Item \ Switching Loss | Original Switching Loss | 1.2 Times the Original Switching Loss |
|---|---|---|
| Optimum average *COP* | 2.3 | 2.1 |
| Worst average *COP* | 2.0 | 1.9 |
| Times reaching optimal value | 27 | 31 |
| Average number of iterations for occurrence of optimal value | 28 | 21 |
| Average execution time (se) | 297.7 | 285.5 |

## 5. Conclusions

This paper proposed a novel bionic heuristic artificial intelligence algorithm, BPA, for solving the excessive random search process of traditional algorithms. BPA integrated various limitations into the translation process of the algorithm for limited on/off status, in order to enhance the robust search capability of the algorithm. After the control strategy of BPA was performed at different time stages, the total power consumption of the system dropped by about 35%, and the *COP* value rose by about 0.6. We note that BPA was successfully used for the refrigerated chamber energy-saving optimization problem, providing a reference direction for enterprise decisions, and it can be used in research works of other related subjects in the future. This paper provides a strategic guideline for enterprise energy-saving and environmental issues, enhances the competitiveness and ability of sustainable operations, substantively helps economic development, and remedies the deficiencies in existing research regarding the energy saving of refrigeration systems.

**Author Contributions:** W.-M.L. designed the algorithm and handled the project as the first author. C.-Y.Y. performed the experiments and conducted simulations. M.-T.T. assisted the project and prepared the manuscript as the corresponding author. C.-S.T. contributed materials and tools. All authors discussed the simulation results and approved the publication.

**Funding:** This research was funded by the Ministry of Science and Technology of the R.O.C. under contract number MOST 105-2221-E-230-012.

**Acknowledgments:** The authors would like to acknowledge the financial support for this work by the Ministry of Science and Technology of the R.O.C. under contract number MOST 105-2221-E-230-012, which is gratefully appreciated.

**Conflicts of Interest:** The authors declare no conflict of interest.

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
