# Peer review of "The Optimized Energy Saving of a Refrigerating Chamber"

_energies, doi:10.3390/en12101887_

Reviewer 1 Report

The paper is interesting reporting new control strategies for climatic chambers. In my opinion the authors have to revise the following aspects:

1) In the Figure 1 and 2 which is the oil and air separator at the suction of the compressor? Air?

2) And the oil separator on the compressor discharge has to allow the return of the oil to the compressor...

3) How is managed the frost at the evaporator wiht the control strategy proposed?

4) Using the on-off valves are there trouble with the lubricant oil of the compressor?

5) The implementation of BPA must be explained better

Author Response

Dear Reviewer:

Thank you for providing us the review’s comments. We have taken care of these precious comments and revised manuscript with the changes clearly identified by a highlighter pen. The point-to-point responses to you are shown below.

1In the Figure 1 and 2 which is the oil and air separator at the suction of the compressor? Air?

 Ans: Located at the suction end of the compressor, it is the accumulator for storing liquid refrigerant. Its role is to separate liquid refrigerant and gaseous refrigerant. Figure 1 and Figure 2 had been re-draw as in pg.3and 4.

2And the oil separator on the compressor discharge has to allow the return of the oil to the compressor...

Ans: Yes, it has. The oil in the oil separator must be return to the compressor by oil pump. However, we did not show it because it is not the focus of this article.

 3How is managed the frost at the evaporator wiht the control strategy proposed?

 Ans: We use a hot-gas bypass valve to deal with frosting problems. However, we did not show it because it is not the focus of this article.

  4Using the on-off valves are there trouble with the lubricant oil of the compressor?

ANS: No, there isn't trouble. There are not trouble with the lubricant oil of the compressor in this system. Because, there is the oil separator on the compressor discharge. The oil in the oil separator will be return to the compressor.

5The implementation of BPA must be explained better.

Ans: The advanced description of BPA had been added with a highlighter pen as in pg.2,7 and 9.

I sincerely hope that we have clarified all your questions. Your assistance is very much appreciated. If you have further questions, please feel free to contact me.

Your assistance is highly appreciated.

Sincerely yours.

Dr. Ming-Tang Tsai

Department of E.E.,

Cheng-Shiu University

Email:[email protected]

Reviewer 2 Report

I this work, authors propose a new control strategy for optimizing energy savings in a specific application that involves several refrigerating chambers. A Binary Proteome Algorithm (BPA) is proposed to solve this problem with a final increment in the average COP. Despite the interest of the manuscript, there is an important lack of information related to the application and the quantification of the energy savings claimed by authors. For example, is difficult to understand how the ON/OFF control strategy of valves can reduce the power consumption of compressors if discharge/suction conditions are fixed and their cubic capacity or/and rotation speed are maintained. Why the compressor is not regulated by a drive or similar in order to adapt its cooling capacity to the cooling demand?. On the other hand, a more detailed operation diagram of the BPA algorithm is necessary since section 3.1 is not integrated with the aim of this work.

Conclusions need to be supported by the results and the energy savings (in terms of annual kW·h) need to be reflected. Unfortunately, this essential information is missing.

Finally, the information about the computer used in the simulations is incomplete. The motherboard brand and model, as well as the RAM memory model and brand, are mandatory if authors would like to show, for example, the operating time of the algorithm 

Author Response

Dear Reviewer:

Thank you for providing us the review’s comments. We have taken care of these precious comments and revised manuscript with the changes clearly identified by a highlighter pen. The point-to-point responses to you are shown below.

1. It is difficult to understand how the ON/OFF control strategy of valves can reduce the power consumption of compressors if discharge/suction conditions are fixed and their cubic capacity or/and rotation speed are maintained. Why the compressor is not regulated by a drive or similar in order to adapt its cooling capacity to the cooling demand?

Ans: Reference:Fu-Sheng Industrial Co., Ltd. Screw Refrigerant Compressor-Instruction manual. 2016.

(a).The cooling ability of compressors is adjusted by 25%, 50%, 75%, and 100% of capacity control. The capacity control of compressors has three electromagnetic valves as following Figure. When the temperature of the refrigerating chamber reaches a prescribed temperature, the electromagnetic valve will be closed so that the refrigerant cannot enter the refrigerating chambers to maintain the prescribed temperature. At the same time, the compressor will be unloaded, ie the cooling capacity will be adjusted according to the temperature of the refrigerating chambers. When the compressor is operated at low cooling capacity, power consumption will decrease. It is independent of discharge/suction conditions, cubic capacity or rotation speed.

(b). The cooling ability of compressors is adjusted from 25% capacity to 100% capacity. Similarly, when the compressor is operated at lower cooling capacity, power consumption will decrease. It is independent of discharge/suction conditions, cubic capacity or rotation speed.

2. A more detailed operation diagram of the BPA algorithm is necessary since section 3.1 is not integrated with the aim of this work.

Ans: The advanced description of BPA had been added with a highlighter pen as in pg.2,7 and 9.

2.   Conclusions need to be supported by the results and the energy savings (in terms of annual kW·h) need to be reflected.

Ans: This paper takes the time stage 11:00~16:00 of the refrigerating chamber in a day as the simulation test example, the five hours are subdivided, and each small time stage is 5 minutes, in order to test the real time efficiency. So, the energy saving calculated in our manuscript is KW. It can reduce the peak load and energy saving through the control strategy of the refrigerating chamber.

3. The information about the computer used in the simulations is incomplete, for example, the operating time of the algorithm.

Ans: The operating time of the algorithm had been added with a highlighter pen as in pg.15 and 16.

I sincerely hope that we have clarified all your questions. Your assistance is very much appreciated. If you have further questions, please feel free to contact me.

Your assistance is highly appreciated.

Sincerely yours.

Dr. Ming-Tang Tsai

Department of E.E.,

Cheng-Shiu University

Email:[email protected]

Reviewer 3 Report

This paper discusses the use of Binary Proteome Algorithm (BPA) for control of refrigerators. The work is interesting, but the paper is poorly written and is frustrating to read.

More detailed comments are given below:

1. There are grammatical errors and unusual phrases found throughout the text. It will be useful to get some advice from people whose first language is English.

2. Abstract: Please include some of the most significant quantitative results in the Abstract.

3. Section 1: Some of the data are quoted without any references.

4. Section 1: Any data for Taiwan’s refrigeration power consumption?

5. Section 1: More discussions are needed for refs [5-14].

6. Section 1: I am not sure [16] and [17] are quoted in the correct format.

7. Section 1: More explanation on why BPA was chosen is needed.

8. Section 2: Use “refrigerant” instead of “coolant”

9. Figure 2: What are the “…………” symbols referring to?

10. Section 3.1: This section is very confusing. I know it is about the BPA, but is there a better way to explain the algorithm to those with no in-depth knowledge about biology?

11. Section 4: Was the work tested experimentally or only in simulation? I somehow cannot get a clear answer to this.

12. Tables 2-6: Please consider to present these in graphs.

13. Figures 9 & 10: They look very different in formats. Please use the same software and template for consistency.

14. Conclusion: Please include some of the most significant quantitative results in the Conclusion.

Author Response

Dear Reviewer:

Thank you for providing us the review’s comments. We have taken care of these precious comments and revised manuscript with the changes clearly identified by a highlighter pen. The point-to-point responses to you are shown below.

1There are grammatical errors and unusual phrases found throughout the text. It will be useful to get some advice from people whose first language is English.

Ans:We appreciate your comment. This manuscript has been revised by a native English speaker to improve the readability.

2Abstract: Please include some of the most significant quantitative results in the Abstract.

Ans: Abstract had been added with a highlighter pen as in pg.1.

3Section 1: Some of the data are quoted without any references.

Ans: References[2] are added in pg.1.

4Section 1: Any data for Taiwan’s refrigeration power consumption?

Ans: Data about Taiwan's refrigeration power consumption can be found on the website of Bureau of Energy, Ministry of Economic Affairs. However, detailed information will not be announced by various manufacturers.

5Section 1: More discussions are needed for refs [5-14].

ANS:More discussions are added with a highlighter pen in pg.2.

6Section 1: I am not sure [16] and [17] are quoted in the correct format.

Ans: The format of [16] and [17] is corrected in pg.18.

7Section 1: More explanation on why BPA was chosen is needed.

 Ans:The more explanation on why BPA had been added with a highlight pen in pg.2.

 8Section 2: Use “refrigerant” instead of “coolant”

 Ans: We had been used “refrigerant” instead of “coolant in this paper.

9Figure 2: What are the “…………” symbols referring to?

 Ans: The symbol is ellipsis of chamber 3 to N-1 which contains electromagnetic valves and expansion valve as in Figure 2.

10Section 3.1: This section is very confusing. I know it is about the BPA, but is there a better way to explain the algorithm to those with no in-depth knowledge about biology?

 Ans:BPA is a new and a bionic algorithm. It refers to an algorithm based on the mechanism of protein production. Knowledge about proteins can be found in basic biology.

 11Section 4: Was the work tested experimentally or only in simulation? I somehow cannot get a clear answer to this.)

 Ans: We use a common refrigerating chamber system for computer simulation. We can get the best energy management strategy of the system through simulation, which can give the manager immediate and feasible decision to reach the goal of energy saving.

12Tables 2-6: Please consider to present these in graphs.

 Ans:Due to the result data is very close, using table presentation is more representative of analysis results than using graph.

13. Figures 9 & 10: They look very different in formats. Please use the same software and template for consistency.

 Ans:Figure 9 is re-draw with the same software(pg.15). (Figure is changed to Figure 10.)

 14Conclusion: Please include some of the most significant quantitative results in the Conclusion

 Ans: The some of the significant quantitative results is added with highlighter pen. (pg.17)

 I sincerely hope that we have clarified all your questions. Your assistance is very much appreciated. If you have further questions, please feel free to contact me.

 Your assistance is highly appreciated.

Sincerely yours.

 Dr. Ming-Tang Tsai

 Department of E.E.,

 Cheng-Shiu University

 Email:[email protected]

Round  2

Reviewer 2 Report

Dear authors. Thank you very much for your accurate answers. I would like to make some references to your comments in order to understand some aspects of the manuscripts.

In the cover letter, the operation of the compressors is presented and described. This information is quite important and it is not reflected in the manuscript. In fact, the compressor type (screw-compressor) is not mentioned in the manuscript so the use of the ON/OFF control strategy is so confusing. You need to include the description made in the cover letter in order to understand that the ON/OFF strategy over climatic chambers is combined with the compressor's electromagnetic valves. Moreover, the model or cubic capacity of the compressor is mandatory.

Abstract. Please, do not include 4 decimals in the results. Instead of 115.9167 kW, use 115.92 kW. On the other hand, kw is wrong, the correct form is kW.

End of Table 6. Similar issue with power consumption.

Author Response

Dear Reviewer:

Thank you for providing us the review’s comments. We have taken care of these precious comments and revised manuscript with the changes clearly identified by a highlighter pen. The point-to-point responses to you are shown as attacted file. This manuscript has been revised by a native English speaker to improve the readability.

I sincerely hope that we have clarified all your questions. Your assistance is very much appreciated. If you have further questions, please feel free to contact me.

 Your assistance is highly appreciated.

Sincerely yours.

Dr. Ming-Tang Tsai

Department of E.E.,

Cheng-Shiu University

Email:[email protected]

Reviewer 3 Report

All my comments have been sufficiently addressed.

Author Response

Dear Reviewer:

We appreciate your comment. This manuscript has been revised by a native English speaker to improve the readability. I sincerely hope that we have clarified all your questions. Your assistance is very much appreciated. If you have further questions, please feel free to contact me.

Your assistance is highly appreciated.

Sincerely yours.

Dr. Ming-Tang Tsai

Department of E.E.,

Cheng-Shiu University

Email:[email protected]
